# Application and Visualization of Fluorescent-Tagged Antiscalants in Electrodialysis Processing of Aqueous Solutions Prone to Gypsum Scale Deposition

**DOI:** 10.3390/membranes12101002

**Published:** 2022-10-16

**Authors:** Violetta Gil, Maxim Oshchepkov, Anastasia Ryabova, Maria Trukhina, Mikhail Porozhnyy, Sergey Tkachenko, Natalia Pismenskaya, Konstantin Popov

**Affiliations:** 1Department of Physical Chemistry, Kuban State University, 149 Stavropolskaya Str., 350040 Krasnodar, Russia; 2Department of Chemical and Pharmaceutical Technologies and Biomedical Pharmaceuticals, Mendeleev University of Chemical Technology of Russia, Miusskaya Sq. 9, 125047 Moscow, Russia; 3Prokhorov General Physics Institute of the Russian Academy of Sciences, Vavilov Str., 38, 119991 Moscow, Russia; 4JSC “Fine Chemicals R&D Centre”, Krasnobogatyrskaya, Str. 42, b 1, 107258 Moscow, Russia

**Keywords:** electrodialysis, scaling, scale inhibition, fluorescent-tagged antiscalants, antiscalant visualization, chronopotentiometry, fluorescent microscopy, gypsum

## Abstract

Membrane scaling is a serious problem in electrodialysis. A widely used technique for controlling scale deposition in water treatment technologies is the application of antiscalants (AS). The present study reports on gypsum scale inhibition in electrodialysis cell by the two novel ASs: fluorescent-tagged bisphosphonate 1-hydroxy-7-(6-methoxy-1,3-dioxo-1Hbenzo[de]isoquinolin-2(3H)-yl)heptane-1,1-diyl-bis(phosphonic acid), HEDP-F and fluorescein-tagged polyacrylate, PAA-F2 (molecular mass 4000 Da) monitored by chronopotentiometry and fluorescent microscopy. It was found that cation-exchange membrane MK-40 scaling is sufficiently reduced by both ASs, used in 10^−6^ mol·dm^−3^ concentrations. PAA-F2 at these concentrations was found to be more efficient than HEDP-F. At the same time, PAA-F2 reveals gypsum crystals’ habit modification, while HEDP-F does not noticeably affect the crystal form of the deposit. The strong auto-luminescence of MK-40 hampers visualization of both PAA-F2 and HEDP-F on the membrane surface. Nevertheless, PAA-F2 is proved to localize partly on the surface of gypsum crystals as a molecular adsorption layer, and to change their crystal habit. Crystal surface coverage by PAA-F2 appears to be nonuniform. Alternatively, HEDP-F localizes on the surface of a deposit tentatively in the form of [Ca-HEDP-F]. The proposed mechanisms of action are formulated and discussed. The application of antiscalants in electrodialysis for membrane scaling mitigation is demonstrated to be very promising.

## 1. Introduction

A scale is a solid layer deposited onto industrial equipment surfaces through a process called scaling. Scale deposition is a problem encountered with water containing ions of sparingly soluble salts, notably calcium carbonate, calcium sulfate, magnesium hydroxide, calcium phosphate, and silica [1]. The most important single factor determining the intensity of scaling is the supersaturation (SS) level of the deposit-forming species. SS conditions are achieved when a solution is concentrated beyond the solubility limits of one or more of its constituents by water evaporation (as in water cooling towers), by the separation of pure water at ambient temperature (as in baromembrane processes) [2], by a local excess of the solubility product of salts, or by a shift in pH due to concentration polarization phenomena (as in electrodialysis processes) [3]. Membrane scaling is a serious problem in both electrodialysis technologies (ED) [4,5,6,7] and reverse osmosis (RO) [8,9,10].

Several methods have been recently developed to combat scale deposition in membrane stacks of electrodialyzers. Some of them are similar to the ones used in other membrane processes, e.g., a preliminary removal of ions, responsible for scale formation, use of chemical reagents, pressure-driven membrane processes, etc. [3,11,12,13]. Besides, there are some techniques, inherent only in ED. Among these, there are: induction of forced electroconvection near the surface of ion-exchange membranes (IEMs) in the desalination compartments of electrodialyzers, which “washes away” the nuclei of scale crystals [4]; the development of IEMs from fundamentally new polymers that have either a hydrophobic [14], or, on the contrary, a hydrophilic surface [15]; creating a small pressure drop between the desalination and concentration compartments to destroy the sediment layer [16]; and using new membrane stack configurations [17,18], where the conditions for scale formation are partially eliminated.

One of the traditional ways to mitigate sediment deposition is the application of electrodialysis reversal [19,20]. This method involves a periodic (once in a few minutes) change in the direction of the electric current and the pumped liquid. At the same time, despite the sophistication of the membrane modules design and the control system of the process of electrodialysis reversal, it is not possible to achieve complete inhibition of scale formation. The simplest method that does not require a radical transformation of the membrane stacks of electrodialyzers is the method of pulsed electric fields (PEF). PEF assumes a short-term (0.001–1 s) electric current switching on (potential drop) and its absence. This method has been tested for scaling mitigation in the dairy industry during the separation of casein and demineralization of whey [21,22], and for decreasing organic fouling in desalination [23] and valuable component extraction from multicomponent solutions [24]. In all cases, it was possible to increase the lifespan of membrane modules without washing them with chemical reagents, and at the same time to achieve high yields of end product without increasing energy costs. However, the reasons for the success achieved (and hence the levers for meaningful process control) are still the subject of discussion, and the selection of optimal current modes is purely of an empirical nature.

A widely used technique for scale deposition control is an antiscalant (AS) injection at sub-stoichiometric amounts (typically 0.5–10 mg/L) [1,2,25]. The most common industrial ASs are represented by phosphonates (aminotris(methylenephosphonic acid), ATMP; 1-hydroxyethane-1,1-bis(phosphonic acid), HEDP; 2-phosphonobutane-1,2,4-tricarboxylic acid, PBTC; etc.) and polycarboxylates (polyacrylates, PAA; polyaspartates, PASP; polyepoxysuccinates, PESA; etc.). Antiscalants are successfully applied in RO technologies according to multiple reviews [9,10,25]. However, the reports on AS applications in ED are sparse and scattered [26,27,28,29,30] due to the more complicated processes in electrodialysis relative to RO. Notably, in some of these studies, antiscalants are used in combination with other methods of scaling mitigation, e.g., electrodialysis reversal [29] and pH adjustment by adding chemical reagents [30]. On the other hand, irrespective of the long-lasting use of AS in RO and in other water treatment technologies, the fundamentals of scale inhibition mechanisms are still incompletely understood, especially from a quantitative approach. Considerable theoretical guidance is available on the nucleation, precipitation, and adherence phenomena involved in scale formation [1], but some issues are still a matter of discussion [5,31]. However, basic information on the nature, amount, and type of interaction induced by AS on scaling processes is scant [2].

Recently the situation has changed due to the possibility to track AS directly in any water treatment process using fluorescent-tagged scale inhibitors [32,33,34]. Indeed, our recent experiments with gypsum scale formation in a laboratory RO setup in the presence of fluorescent-tagged bisphosphonate 1-hydroxy-7-(6-methoxy-1,3-dioxo-1Hbenzo[de]isoquinolin-2(3H)-yl)heptane-1,1-diyl-bis(phosphonic acid), HEDP-F (H_4_hedp-F) [32] and of fluorescent 1,8-naphthalimide-tagged polyacrylate (PAA-F1) [33] revealed a paradoxical effect: an antiscalant does not show any kind of interaction with gypsum, but provides nevertheless retardation of the corresponding deposit formation.

Particularly, HEDP-F was found to form solid fluorescent particles of [Ca-HEDP-F] in the RO experiment long before the gypsum supersaturation point was reached. At the same time, when gypsum crystals started to stand out, they revealed no traces of HEDP-F; neither on their surface, nor inside the CaSO_4_·2H_2_O phase [32]. In this way, antiscalant was evidently disabled as the [Ca-HEDP-F] species before the nucleation of gypsum had started, but it still provided retardation of deposit formation. The same effect was observed during gypsum deposition in an RO facility in the presence of PAA-F1 (see [33] and references there). Besides, in a static experiment at ambient temperature (supersaturation is achieved by mixing Na_2_SO_4_ and CaCl_2_ solutions), we have detected a similar result: classical stick-like crystals of gypsum mixed with spherical [Ca-HEDP-F] particles. However, when a mixture of Na_2_SO_4_ and CaCl_2_ solutions was heated at 70 °C for 24 hours according to NACE Standard TM0374-2007 and then cooled, the result was very different. We have observed [Ca-HEDP-F] particles inside the gypsum crystals and some traces of a HEDP-F molecular layer on the CaSO_4_·2H_2_O crystal surface (see [33] and references there). This indicates that: (i) in particular cases [Ca-HEDP-F] particles are capable to act as crystallization centers of gypsum; and (ii) the mechanisms of scale inhibition may vary significantly depending on temperature, supersaturation degree, and antiscalant concentration, etc. Therefore, it was of special interest to study gypsum deposit formation in electrodialysis, where supersaturation is achieved via a slow electromigration of calcium ions to the Na_2_SO_4_ solution. This situation is rather different from that one in earlier RO experiments run by us [32,33].

Thus, the objective of this study was to investigate the activity of the two new scale inhibitors in electrodialysis of the solutions prone to gypsum deposition and to estimate their possible localization in the membrane stack. We used fluorescent-tagged bisphosphonate HEDP-F and fluorescein-tagged polyacrylate PAA-F2. The first one is a fluorescent analogue of industrial scale inhibitor HEDP, while the other one models the behavior of another industrial antiscalant-polyacrylate PAA.

In order to accomplish the objective, calcium sulfate deposition was studied in an electrodialysis cell by means of chronopotentiometry, followed by optical microscopy, scanning electron microscopy (SEM), and fluorescent microscopy (FM) analysis of membrane surface. Gypsum was chosen as a model deposit, since its sedimentation in electrodialysis is quite a problem and is widely discussed in the literature [35,36,37]. In addition, gypsum is a rather convenient object of research with well-known properties. To the best of our knowledge, our study is the first attempt to apply fluorescent antiscalants in an electrodialysis system worldwide.

## 2. Materials and Methods

### 2.1. Reagents

Commercial-reagent-grade NaCl, Na_2_SO_4_, CaCl_2_, and deionized water were used for the preparation of the 0.04 mol·dm^−3^ individual solutions. PAA-F2 and HEDP-F were synthesized as described elsewhere [38,39]. Figure 1 shows the structure of these antiscalants. HEDP-F exhibits the blue light emission (fluorescence band at 460 nm), while PAA-F2 reveals green fluorescence (fluorescence band at 505 nm). It should be noted, that though 1,8-naphthalimide-based antiscalants provide fluorescence in the blue spectral region, the fluorescent channel can be assigned any pseudo-color in the digital image when acquired during laser scanning microscopy. In the present study, the fluorescent channel assigned for HEDP-F was the green pseudo-color because the dynamic range of human color perception for green is much wider than for blue. Thus, the original blue emission of HEDP-F was changed to an artificial green pseudo-color.

### 2.2. Electrodialysis Experimental Setup

Electrodialysis experiments were run in an experimental setup shown in Figure 2. The setup includes the four-compartment flow electrodialysis cell (1). Compartments were formed by the cation-exchange membrane (CEM) MK-40 to be studied and by two auxiliary anion-exchange membranes (AEM) MA-41. Both MK-40 and MA-41 membranes are manufactured and supplied by Shchekinoazot Ltd., Russia. The membranes were manufactured by hot-rolling of ground ion-exchange resins KU-2-8 (MK-40) and AV-17-8 (MA-41) and a low-pressure polyethylene powder. Then, a nylon reinforcing cloth was introduced into them by the hot-pressing method [40]. The grain size of the ion-exchange resin ranged from 5 to 50 µm. The ion-exchange matrix of all these membranes was made of a styrene-divinylbenzene copolymer. CEM MK-40 contains fixed sulfonate groups; the fixed groups of the MA-41 anion-exchange membrane were represented by quaternary amines and a small amount of secondary and tertiary amines [40]. The intermembrane distance was 6.6 mm, the polarized membrane area was 2 × 2 cm^2^.

In all runs the 0.04 mol·dm^−3^ CaCl_2_ solution was constantly fed from tank 2 into the desalination compartment (DC), while 0.04 mol·dm^−3^ Na_2_SO_4_ solution was delivered in the parallel way from tank 3 to the concentration compartment (CC). At such solution concentrations, the CaSO_4_·2H_2_O deposit was formed in the CC at a relatively high rate due to the transfer of Ca^2+^ ions across the cation-exchange membrane and a significant increase in their concentration near the MK-40 surface facing the concentration compartment (Figure 3a). At the same time the 0.04 mol·dm^−3^ NaCl solution was pumped through the electrode compartments. The average linear flow velocity was adjusted to 0.4 cm/s in all compartments.

The potential drop across the membrane was determined using two measuring probes, Figure 2 (silver chloride electrodes (6) connected to Luggin’s capillaries (5)). The potential drop was measured using the Keithley 2010 multimeter (Keithley Instruments, Inc., Cleveland, OH, USA) (7). The electric current was set by the programmable power supply Keithley 2200-60-2 (Keithley Instruments, Inc., USA) (8) and supplied to the cell (1). The pH of the solution in the concentration compartment was measured in a flow cell with a combined pH electrode (9) using the Expert-001 pH meter (Econiks-Expert, Ltd., Moscow, Russia) (10). All the experiments were run at 25 ± 1 °C.

An assessment of antiscalant activity was performed by a comparison of chronopotentiograms, registered in the blank experiment run without scale inhibitors and in the presence of 10^−6^ mol·dm^−3^ of either PAA-F2 or HEDP-F, added to the concentration compartment. Bearing in mind the molecular masses of PAA-F2 (4000 Da) and HEDP-F (559.44 Da), these correspond to the 4 mg·dm^−3^ (PAA-F2) and 0.6 mg·dm^−3^ (HEDP-F) concentrations.

### 2.3. Membrane Scaling Characterization

After the 3-hour electrodialysis experiments the cation-exchange membranes were removed, and their surfaces facing the concentration compartment were analyzed by the optical microscope SOPTOP CX40M (Ningbo Sunny Instruments Co., Ltd., Yuyao, China). After that, the membranes were rinsed by deionized water and dried at ambient temperature. Then, the side covered by gypsum was studied with scanning electron microscopy. The sample examination by SEM (Hitachi TM 3030, Hitachi, Tokyo, Japan) was done at 15 kV accelerating voltage in Charge-Up Reduction Mode and 4.1 mm working distance. Energy dispersive X-ray spectroscopy (EDS) data was processed using QUANTAX 70 software. System calibration was performed using copper standard. EDS data were acquired in Spot Mode and Mapping mode. Elements coloring was used in the automatic mode. Spot Mode was used for quantitative element analysis, and Mapping mode for elements distribution localization.

Fluorescent microscopy measurements were run with a laser scanning confocal microscope LSM-710-NLO (Carl Zeiss Microscopy, Jena, Germany), at ×20 Plan-Apochromat objective (NA = 0.8). The fluorescence of the HEDP-F and PAA-F2 was recorded in the wavelength range of 470–600 nm, when excited by a laser with a wavelength of 488 nm. The reflected scanning laser light was recorded as an additional channel. The 3D fluorescence images were recorded with a step 1 µm along the Z axis.

## 3. Results

### 3.1. Chronopotentiometry

To analyze the process of deposit formation in the ED systems, 3-hour experiments on the electrodialysis processing of solutions were carried out under conditions of the constant overlimiting current *i* = 10 mA/cm^2^. The reduced value of the potential drop, ∆*φ*′, defined as the difference between the total potential drop and the ohmic potential drop that occurs in a non-polarized membrane system instantly after the current is turned on [41] was used for chronopotentiogram (ChPs) plotting. In the (quasi)stationary state, the ChP of the blank experiment had periods of noticeable growth and a decrease in the values of the potential drop (Figure 4a, curve I). The periods of growth are associated with the formation of deposit crystal structures on the membrane surface, which prevent the transfer of ions and, accordingly, lead to an increase in the system resistance [4,42]. The periods of potential drop decrease may be associated with these structures partially washing off (when the crucial mass of the deposit is accumulated) under the influence of the hydrodynamic factor caused by the forced flow of the solution. However, the membrane surface becomes almost completely covered by the deposit layer after the three-hour blank experiment; see Figure 2b,c and Figure 5a. Notably, the formation of deposit crystals occurred primarily on the grains of the ion-exchange resin, protruding from the polyethylene surface [43]. Further, under favorable conditions, the deposit spread over the rest of the membrane surface.

In the presence of PAA-F2, the significantly smaller potential drop values were registered (Figure 4a, curve II) and a negligible amount of scale was deposited, Figure 5b. Thus, PAA-F2 was capable of suppressing scaling almost completely during 3-hour experiments on the electrodialysis processing of the solutions. HEDP-F demonstrated a lower antiscaling efficiency relative to PAA-F2. However, it revealed a less prominent increase in ∆*φ*′ (Figure 4a, curve III) and provided a diminished coverage of the membrane surface by the gypsum layer (Figure 5c) in comparison with the blank run.

The pH of the solution pumped through the concentration compartment increased gradually in all three systems (Figure 4b, Table 1) due to the more intense reaction of water splitting at the AEM surface facing the cathode compartment than the similar reaction at the surface of the CEM facing the desalination compartment. Water splitting takes place as a result of water molecules undergoing protonation-deprotonation reactions with the participation of the fixed ion-exchange membrane groups acting as catalytic centers. The catalytic activity of the fixed groups of the AEM MA-41 is quite high relative to that of one of the fixed groups of the CEM MK-40 [44,45]. A slower increase in pH in systems containing antiscalants (Figure 4b, curves II, III) is apparently caused by the participation of phosphonate (HEDP-F) and carboxylate (PAA-F2) groups of antiscalants in protonation-deprotonation reactions. Another factor that can potentially influence the pH change in the concentration compartment is the gypsum deposit itself. It creates a layer on the CEM surface facing the CC that provides an additional barrier to block the transport of H^+^ ions through the opposite side of the membrane.

Both antiscalants demonstrate the ability to suppress gypsum scale formation on the membrane surface, and to enhance the efficiency of electrodialysis process. The latter might be provided by several different mechanisms: (i) the reduction of area covered by the deposit and (ii) a decrease in scale layer density.

The reduction of the deposit area can occur either by the formation of fewer gypsum crystals over a certain period of time, or by the reduced binding of gypsum crystals to the membrane surface and the subsequent removal of part of them by the solution flow in the electrodialysis cell. It should be pointed out here that the sub-stoichiometric antiscalants are not capable of preventing the scale formation phenomenon [2]. The exact amount of inorganic sparingly soluble salt that exceeds the thermodynamic solubility level will inevitably precipitate as a solid. Thus, the sub-stoichiometric antiscalants are merely able to postpone the moment of deposition. At the same time, the retardation of scale deposition and the corresponding prolongation of the membrane lifespan may become of key importance for electrodialysis. There are also several possible mechanisms of such retardation depending on the scale nucleation habit [1,33]. First of all, antiscalants may block in several ways the nucleation step of a sparingly soluble salt in the bulk aqueous solution. On the other hand, they can also block the nucleation centers on the membrane surface. Besides, ASs may adsorb onto the surface of the already formed small crystals of inorganic salt, and thus slow down their further growth and aggregation. Any partial refinement of processes that govern the observed scale inhibition in electrodialysis seems to be valuable.

### 3.2. SEM and FM Analysis of Membranes

In order to acquire at least a partial insight into the antiscaling activity of PAA-F2 and HEDP-F in electrodialysis cells, SEM and FM analyses were performed. A pristine cation-exchange membrane MK-40 has a nonhomogeneous surface structure; see Figure 6a. Unfortunately, its material possesses intensive auto-luminescence in a broad spectrum, ranging from 460 to 650 nm; see Figure 6b. This circumstance significantly complicates the localization of fluorescent-tagged PAA-F2 and HEDP-F against the membrane background. On the other hand, localization of gypsum particles as dark objects in the images becomes possible. Thus, in the present study any 3D fluorescent image (Figure 6d) is accompanied by a corresponding 3D image of the same membrane sector, obtained in the reflected light, presented in a lilac pseudo-color (Figure 6c), and by a combination of reflected light and fluorescence emission channels (Figure 6b). The 3D image in the reflected light channel characterizes the same area of the membrane surface that emits fluorescence, but without any special indication of fluorescent fragments. A combination of reflected light and fluorescence emission channels provides an opportunity to localize nonfluorescent domains of membranes and particles of deposit as lilac areas, whereby the fluorescent area is represented only by fragments of the membrane that are not covered by sediment and are able to emit fluorescence. Fluorescent microscopy analysis reveals a cellular structure of the pristine membrane with the domain’s mean size ranging from 10 to 50 µm, which corresponds to the grains of the ion-exchange resin; see Figure 6b.

MK-40 membrane’s surface becomes completely covered by calcium sulfate scale after the blank experiment; see Figure 5a and Figure 7. SEM images reveal typical stick-like crystals of gypsum along with a shapeless array of calcium sulfate scale; see Figure 7a. Accordingly, the fluorescent domains of the membrane become invisible under the deposit layer; see Figure 7b. Only a few fluorescent spots were detected; see Figure 7b,d. At the same time, the scale mass looks rather loose, with a lot of pores (Figure 7c), capable of providing electrolyte ion migration.

The application of PAA-F2 antiscalant provides a dramatic change in membrane surface appearance compared to the blank run; see Figure 5b and Figure 8. SEM and optical microscopy indicate sufficient areas of deposit-free membrane surface; see Figure 5b and Figure 8a. At the same time the morphology of sediment is also changed. The loose deposit matter has vanished. The scale mass is represented by compact aggregates of well-shaped stick-like gypsum crystals, tightly attached to one another; see Figure 8a. Notably, these crystals are less elongated than those deposited in the blank run; see Figure 7a. Thus, in addition to the reduction in deposit mass, the antiscalant also provides some changes to the crystal habit. Fluorescent images (Figure 8b,d) look very similar to those of the pristine membrane; see Figure 6b,d. At first glance, differentiation between the PAA-F2 fluorescence and that of one of the membrane domains is hardly possible in this case. However, a thorough examination of images 8b and 8c can provide some valuable information; see Figure 9.

A comparison of Figure 9a,b makes it possible to identify small separate gypsum crystals on the membrane surface. These are marked with circles. The lowest particle in the left corner of the 9d image demonstrates that PAA-F2 partly and nonuniformly covers the surface of the gypsum crystal. The same situation is observed for all other gypsum crystals, marked with circles in Figure 9b,d. Unfortunately, it was not possible to determine whether PAA-F2 molecules are also present on the membrane surface domains. Nevertheless, evidently PAA-F2: (i) reduced scale deposition; (ii) localized partly on the surface of the deposit as a molecular adsorption layer; and (iii) provided gypsum crystals with habit modification.

The application of HEDP-F antiscalant did not provide drastic changes in the character of deposition at the membrane surface, compared to PAA-F2. Indeed, the optical microscope indicates small areas of deposit-free membrane surface, which, however, exceed those observed in the blank run; see Figure 5. SEM images reveal the presence of both stick-like crystals and the loose solid deposit; see Figure 10a. However, in contrast to the blank run, Figure 7b, some aggregates of antiscalant-containing small particles are visible on the deposit surface; see Figure 10b,d.

In order to refine this fact, a SEM-EDS multielement distribution was performed with the emphasis on phosphorus, present in HEDP-F molecules; see Figure 11. Indeed, SEM-EDS maps indicate that phosphorus is less abundant throughout the sample than calcium and sulfur, and it is distributed all over the whole gypsum layer. At the same time, the fluorescent aggregates, detected by FM on the deposit layer surface (Figure 10b) look very similar to those found by us previously in RO experiments with gypsum [32,33]. In the latter case such aggregates have been attributed to the [Ca-HEDP-F] separate phase.

On the whole, HEDP-F: (i) reduced scale deposition, but less efficiently than PAA-F2 at the concentration used in the present study; (ii) localized in the scale mass; (iii) localized on the surface of deposit tentatively in the form of [Ca-HEDP-F]; and (iv) did not provide noticeable gypsum crystal habit modification.

## 4. Discussion

Recent reports on antiscalants’ application in electrodialysis either do not consider the mechanism of inhibition at all [27,28,29,30], or suppose that the polymer blocks the nucleation centers of deposit without corresponding experimental approval, and without specification of the nature of these nucleation centers [26]. So far, no reports on antiscalant location in electrodialysis cells with deposit have been published. Generally, it is supposed for all water treatment technologies, that the scale inhibitory mechanisms include [1,2,9,10,25,27]: (i) threshold inhibition (a substoichiometric amount of inhibitor slows down crystal growth or precipitation due to hindering of the homogeneous nucleation step); (ii) chelation (e.g., selective sequestration of some ions by chelation with antiscalant making them unavailable for scaling); (iii) dispersion (adsorption of antiscalants on crystals in suspension to impart the charge which separates them, thus preventing scale creation, or electrostatic and/or steric interactions preventing deposition/sedimentation); (iv) crystal structure modification (change in the orderly growth of crystalline scale making it deformed and slower growing, also by the oriented selective preferable adsorption on crystal active growing sites). However, there are no reports where these mechanisms are differentiated and an increment of each one in total inhibition effect is estimated. In this context, any refinement of a particular mechanism in a particular situation becomes valuable.

In our case, both HEDP-F and PAA-F2 are capable to bind calcium cations. However, being used in substoichiometric amounts, they are not capable of affecting gypsum solubility. Moreover, a reversed order of complexation and inhibition abilities were found in the present study: a chelating agent with the lowest complex forming ability towards calcium ions (PAA-F2) provides a higher antiscaling efficacy relative to HEDP-F, which forms more stable complexes, but reveals a lower antiscaling effect. A similar effect was registered by us earlier for HEDP and PAA [46]. Thus, recent results give an additional argument that thermodynamic stability of complexes is not decisive in the process of scale inhibition.

Our previous studies of HEDP and PAA under similar conditions indicated that the values of zeta-potentials of gypsum particles were not high enough to provide electrostatic repulsion that was capable of resisting particle aggregation [46]. Therefore, this mechanism was unlikely to contribute to the inhibition process.

In the present report, HEDP-F was not adsorbed by the deposit’s surface, but formed a separate phase Ca-HEDP-F, similar to that one found in [32,39]. Hence, crystal structure modification was not the case in this particular situation. However, PAA-F2, unlike PAA-F1 in [33] and HEDP-F in the present study, becomes adsorbed nonuniformly on the surface of gypsum crystals, changing their crystal habit. This in turn may contribute to a better electric current conductivity through the deposit layer.

Thus, the threshold inhibition seems to become the predominant effect, responsible for inhibition on the whole. However, our latest studies, summarized in [47], reveal that scale formation in the bulk aqueous phase has a heterogeneous rather than a homogeneous nature. Recent publications on gypsum bulk crystallization indicate the crucial role of natural nanoimpurities with a particle size < 200 nm (nanodust) in solid phase formation kinetics [32,48,49,50,51]. At the same time, it was shown that such nanoimpurities are an unavoidable factor in any pure chemicals and solvents, to say nothing of industrial waters [50,51]. Indeed, foreign solid particles sized over 100 nm are certainly found in any reagent-grade chemicals in amounts from 10^2^ to ≥10^5^ units per 1 mL (from 10^5^ to ≥10^8^ per dm^3^), including water for ion chromatography (Sigma–Aldrich, St. Louis, MO, USA), KCl (Sigma–Aldrich) and HNO_3_ for microelectronics (Avantor Performance Materials, Ltd., Radnor, PA, USA) [50]. Moreover, the number of nanodust particles with a particle size under 100 nm is higher by at least one order of magnitude [51]. The application of fluorescent-tagged antiscalants has shown that the spontaneous bulk nucleation of gypsum in supersaturated aqueous solutions takes place exactly on the surface of such particles serving as crystallization centers [33,47]. Herewith, the major role of the antiscalant is to block the surfaces of these nanoimpurities rather than the surface of the gypsum nuclei. The most paradoxical effect was detected for HEDP-F in both static [33,47] and RO [32,39] experiments. In both cases, HEDP-F formed a separate phase [Ca-HEDP-F], while gypsum crystals grew in a parallel way without any traces of antiscalant on their surface. The similar behavior of HEDP-F was also registered in the present study; see Figure 10b. One can see that HEDP-F was localized on the surface of the gypsum deposit in the form of separate green formations, which were very similar to those observed in [39] and [32]. Meanwhile, the bulk gypsum sediment (scale density) did not seem to differ significantly from that one, obtained in a blank run; see Figure 7c and Figure 10c. The differences are related mainly to the sediment amount; see Figure 5a,c.

Thus, it is highly reasonable to propose that HEDP-F blocks initially the sufficient part of the solid impurities, present in the concentration compartment. The rest of these impurities are likely to be occupied by sulfate ions. Then, when Ca^2+^ ions enter the CC, they form [CaSO_4_]^0^ complexes in a liquid phase, and get adsorbed on solid impurities (SI), covered either by HEDP-F ([SI·nHEDP-F] particles), or by SO_4_^2−^/[CaSO_4_]^0^ species ({SI·nSO_4_^2−^·z[CaSO_4_]^0^} particles):mCa^2+^ + [SI·nHEDP-F]  ⟶  [SI·mCa·nHEDP-F](1)
nCa^2+^ + {SI·nSO_4_^2−^·z[CaSO_4_]^0^}    ⟶    [SI·(n+z)CaSO_4aq_](2)

Growth and association of [SI(m+z)·CaSO_4_aq] results in gypsum crystal formation, while a similar parallel process for [SI·mCa·nHEDP-F] leads finally to Ca-HEDP-F separate phase isolation. As the number of gypsum nucleation centers in the presence of the antiscalant is much lower than in the blank experiment (due to their blockage by HEDP-F), scale formation slows down.

PAA-F2 is supposed to mask nanodust particles more efficiently than HEDP-F. Thus, much less [SI·mCaSO_4_aq] particles are formed in the CC relative to the case of HEDP-F. This results in a minor deposition of gypsum during the 3-hour experiment. Evidently, [SI·mCa·nPAA-F2] particles and their aggregates are also formed. Unfortunately, these are not visible on the membrane surface. At the same time, PAA-F2 is also found on the surface of gypsum crystals, which become sufficiently distorted in comparison with the blank run. Therefore, it is reasonable to suppose a two-component type of activity of PAA-F2: (i) reduction in the number of gypsum nucleation centers (nanodust blockage), and (ii) a change in deposit texture, which might become more porous, looser and more permeable for electric current.

On the whole, antiscalant visualization definitely provides valuable information on scale–antiscalant interactions in electrodialysis relative to RO processes [32,33,39] and static experiments [39,47], as reported earlier. We suppose that a substitution of fluorescent fragments in PAA-F1 and PAA-F2 for the red-light emitters would give an opportunity to detect antiscalants better on the ion-exchange membrane surfaces.

## 5. Conclusions

Two novel fluorescent-tagged antiscalants PAA-F2 and HEDP-F have been tested in an electrodialysis cell with an objective to diminish gypsum scaling on the surface of the cation-exchange membrane MK-40. Both scale inhibitors, taken at the 10^−6^ mol·dm^−3^ concentration, provided a sufficient reduction of deposit formation during 3-h electrodialysis experiments. PAA-F2 at this concentration was found to be more efficient than HEDP-F. At the same time, PAA-F2 was found to change the habit of gypsum crystals, while HEDP-F did not noticeably affect the form of deposit crystals.

The strong auto-luminescence of MK-40 hampers visualization of both PAA-F2 and HEDP-F on the membrane surface. Nevertheless, PAA-F2 was proved to localize partly on the surface of gypsum crystals as a molecular adsorption layer. Crystal surface coverage by PAA-F2 appeared to be nonuniform. Alternatively, HEDP-F localized on the surface of deposit tentatively in the form of [Ca-HEDP-F].

## Figures and Tables

**Figure 1 membranes-12-01002-f001:**
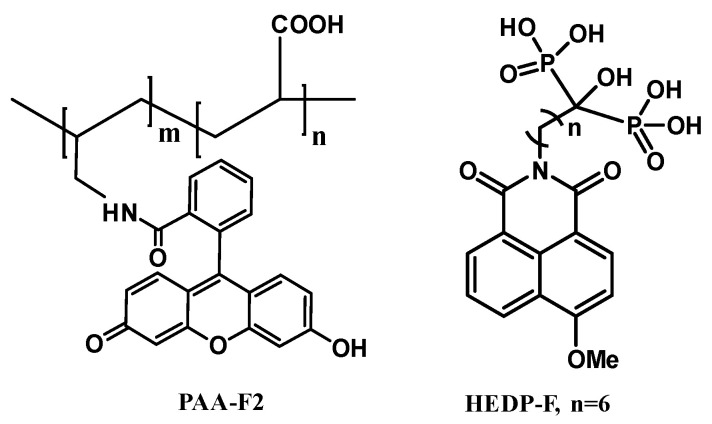
PAA-F2 and HEDP-F molecular structures.

**Figure 2 membranes-12-01002-f002:**
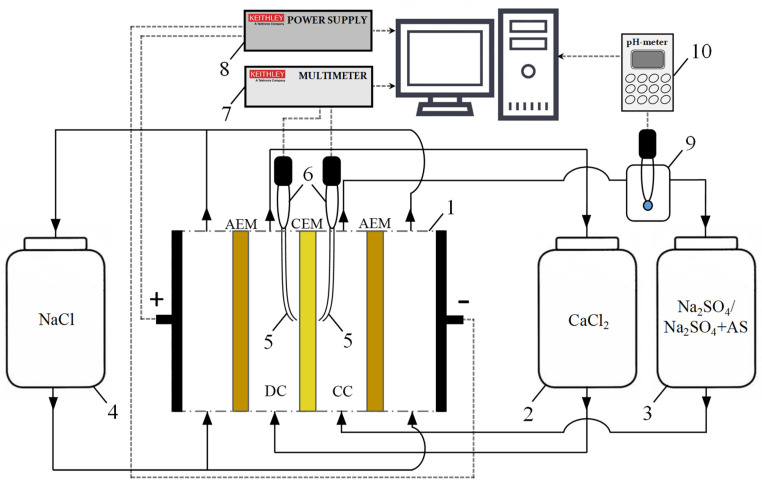
Principal scheme of the experimental setup: electrodialysis cell (1), consisting of one concentration, one desalination and two electrode compartments; solution tanks (2, 3, 4); Luggin’s capillaries (5), connected with silver chloride electrodes (6); Keithley 2010 multimeter (7); Keithley 2200-60-2 power supply (8); and flow pass cell with pH combined electrode (9), connected with pH-meter Expert 001 (10).

**Figure 3 membranes-12-01002-f003:**
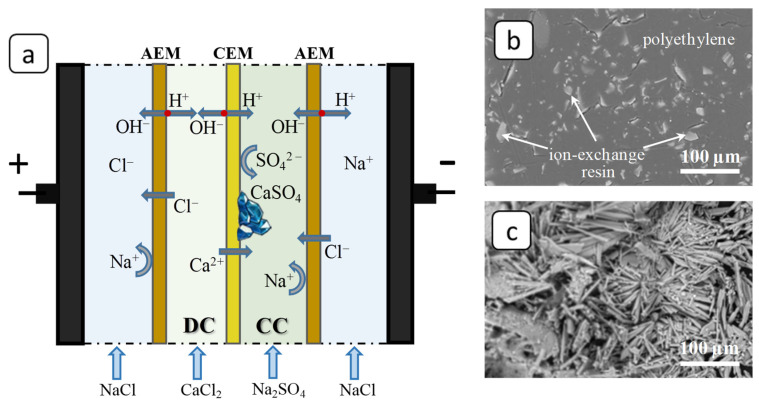
The scheme of ion fluxes in the electrodialysis cell (**a**) and SEM images of the CEM surface facing the concentration compartment before (**b**) and after (**c**) the blank experiment.

**Figure 4 membranes-12-01002-f004:**
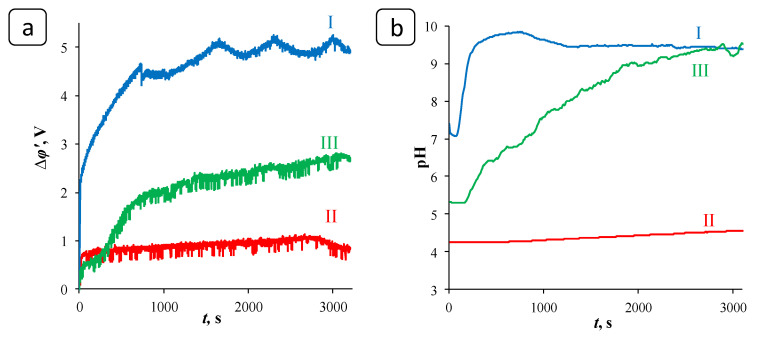
Fragments of chronopotentiograms (**a**) and time dependences of the pH value in the concentration compartment (**b**) for the blank experiment (I) and for experiments in the presence of PAA-F2 (II) and HEDP-F (III) in the constant current mode at *i* = 10 mA·cm^−2^.

**Figure 5 membranes-12-01002-f005:**
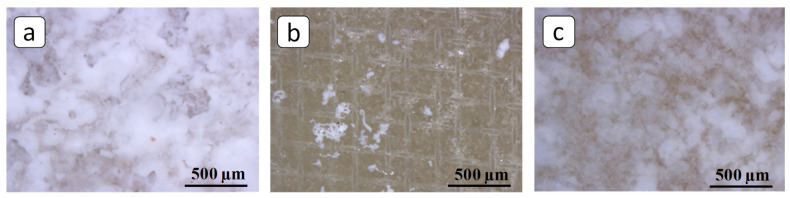
Optical images of the membrane surfaces facing the concentration compartment after the 3-hour experiments on the electrodialysis processing of the solutions: blank experiment (**a**), experiment in the presence of PAA-F2 (**b**) and HEDP-F (**c**).

**Figure 6 membranes-12-01002-f006:**
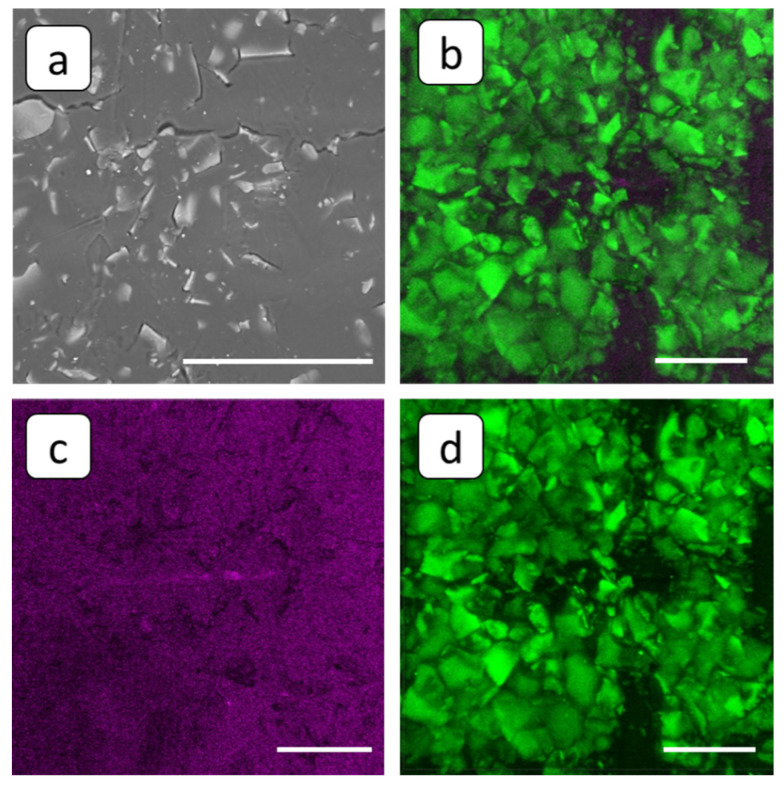
SEM (**a**) and FM 3D images (**b**–**d**) of the pristine cation-exchange membrane MK-40, superposition of fluorescent and reflected light (**b**), and the same view, separately presented in reflected (**c**) and fluorescent (**d**) light. Scale bar: 100 µm.

**Figure 7 membranes-12-01002-f007:**
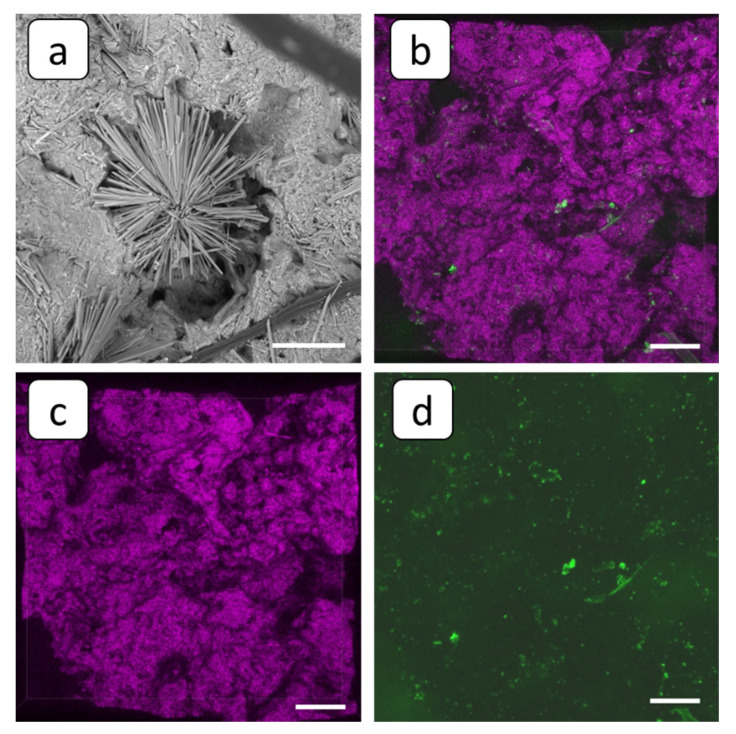
SEM (**a**) and FM 3D images (**b**–**d**) of the cation-exchange membrane MK-40 after the blank experiment, superposition of fluorescent and reflected light (**b**), and the same view in reflected (**c**) and fluorescent (**d**) light. Scale bar: 100 µm.

**Figure 8 membranes-12-01002-f008:**
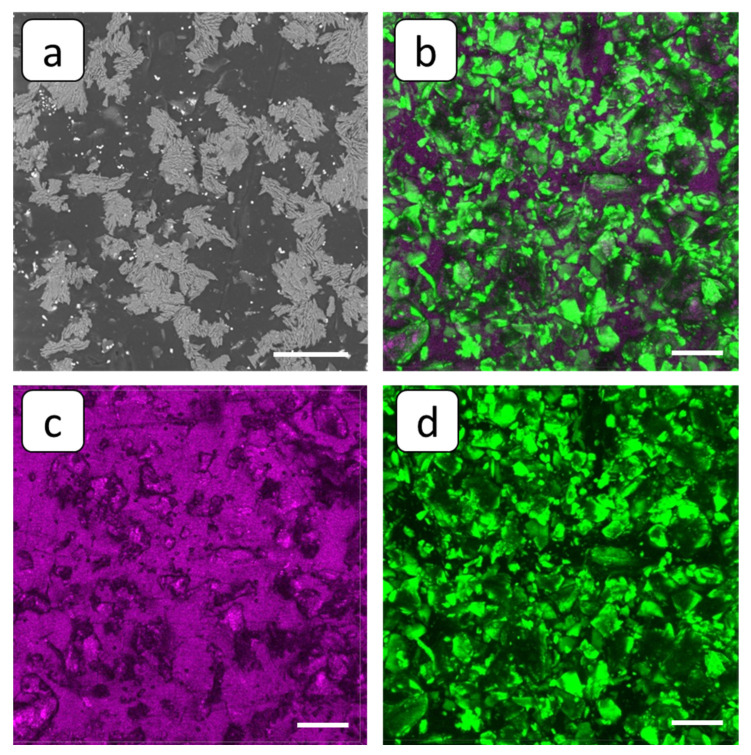
SEM (**a**) and FM 3D images (**b**–**d**) of the cation-exchange membrane MK-40 after the experiment with PAA-F2, superposition of fluorescent and reflected light (**b**), and the same view in reflected (**c**) and fluorescent (**d**) light. Scale bar: 100 µm.

**Figure 9 membranes-12-01002-f009:**
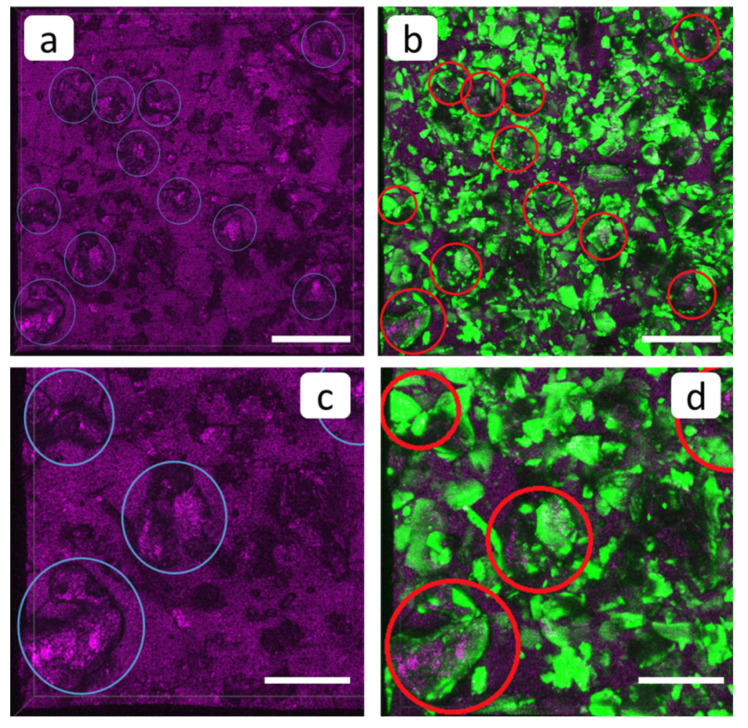
FM 3D images of the cation-exchange membrane MK-40 observed after the experiment run in the presence of PAA-F2, reflected light channel (**a**,**c**) and as a superposition of fluorescent and reflected light (**b**,**d**). Scale bar: 100 µm (**a**,**b**) and 50 µm (**c**,**d**).

**Figure 10 membranes-12-01002-f010:**
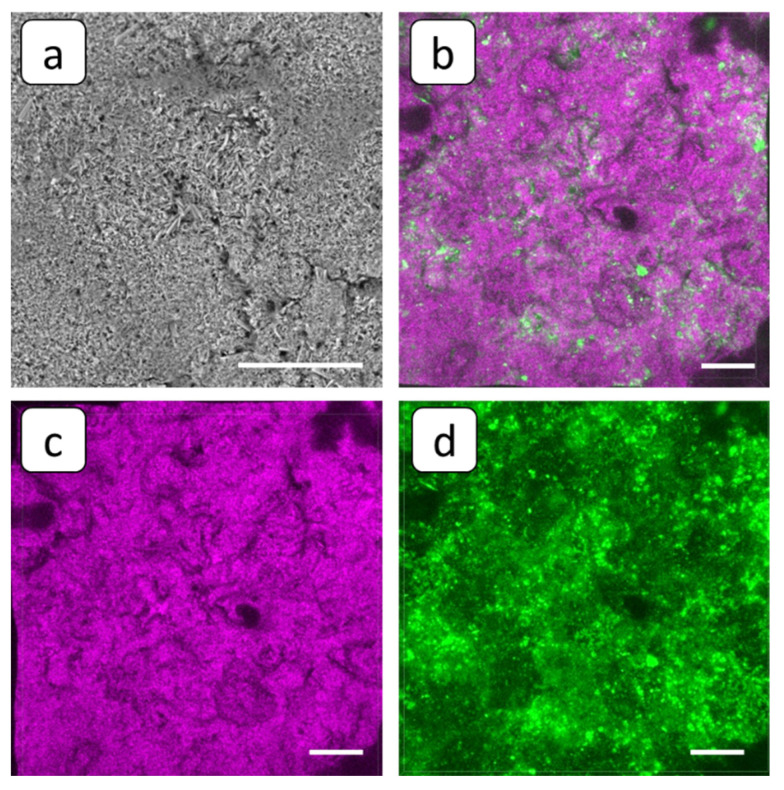
SEM (**a**) and FM 3D images (**b**–**d**) of the cation-exchange membrane MK-40 after the experiment with run in the presence of HEDP-F, superposition of fluorescent and reflected light (**b**), and the same view in reflected (**c**) and fluorescent (**d**) light. Scale bar: 100 µm.

**Figure 11 membranes-12-01002-f011:**
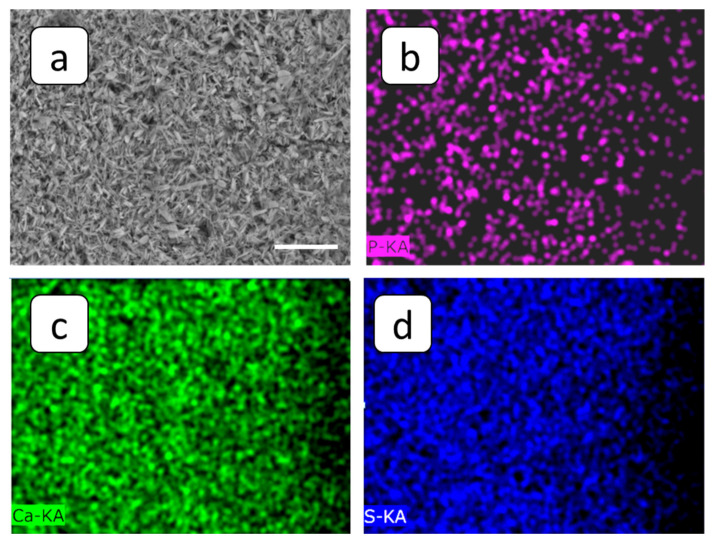
Images of the cation-exchange membrane MK-40 after the experiment run in the presence of HEDP-F: SEM image (**a**) and SEM-EDS maps of phosphorus (**b**) calcium (**c**) and sulfur (**d**). Scale bar: 60 µm.

**Table 1 membranes-12-01002-t001:** pH values in the concentration compartment.

Measurement Time, h	pH		
	Blank	In the Presence of PAA-F2	In the Presence of HEDP-F
0, Initial pH	7.40	4.25	5.31
3, Final pH	8.85	5.38	9.66

## Data Availability

Not applicable.

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
