# Peer review of "Application and Visualization of Fluorescent-Tagged Antiscalants in Electrodialysis Processing of Aqueous Solutions Prone to Gypsum Scale Deposition"

_membranes, 2022, doi:10.3390/membranes12101002_

Round 1

Reviewer 1 Report

The article is devoted to the study of antiscalants in electrodialysis processing, which is of great practical importance. Two new antiscalants are being investigated by chronopotentiometry and fluorescent microscopy. The article describes in detail the experiments carried out, all the conclusions made are substantiated. At the same time, it should be noted that it is necessary to describe in more detail how to interpret the results of reflected light images, for example, Fig. 6c.

Author Response

Thank you very much indeed for this comment. The corresponding details are now added at page 8: 

In order to acquire at least a partial insight into antiscaling activity of PAA-F2 and HEDP-F in electrodialysis cells, SEM and FM analysis were performed. A pristine cation-exchange membrane MK-40 has a nonhomogeneous surface structure, Figure 6a. Unfortunately, its material possesses intensive auto-luminescence, in a broad spectrum, ranging from 460 to 650 nm, Figure 6b. This circumstance significantly complicates localization of fluorescent-tagged PAA-F2 and HEDP-F against membrane background. On the other hand, localization of gypsum particles as dark objects in the images becomes possible. Thus, in the present study any 3D fluorescent image (Figure 6d) is accompanied by a corresponding 3D image of the same membrane sector, obtained in the reflected light, presented in lilac pseudo-color (Figure 6c), and by a combination of reflected light and fluorescence emission channels (Figure 6b). The 3D image in the reflected light channel characterizes the same area of the membrane surface that emits fluorescence, but without special indication of fluorescent fragments, whereby the fluorescent area is represented only by fragments of the membrane that are not covered by sediment and are able to emit fluorescence. A combination of reflected light and fluorescence emission channels gives an opportunity to localize nonfluorescent domains of membrane and particles of deposit as lilac areas. Fluorescent microscopy analysis reveals a cellular structure of virgin membrane with domain’s mean size ranging from 10 to 50 µm, which corresponds to the grains of the ion-exchange resin, Figure 6b.

Reviewer 2 Report

The paper evaluates the efficiency of two known antiscalants in electrodialysis applications and intends to understand at least the principles of anti-scaling mechanisms of both chemicals. The manuscript is well structured, has an excellent scientific approach, and is very easy to follow for readers. My only recommendation is to perform a general revision for typos and, especially excessive use of commas. Besides, in Figures 2 and 3, the readers find asterisks that this reviewer is unsure of their purposes.

The discussion presented in lines 180-183 aroused the curiosity of this reviewer about the evident difference in the efficiency of both antiscalants. Later, in 307-327, the authors added some discussions about the anti-scaling mechanisms which seem extremely reasonable. Nevertheless, this reviewer would like to bring a few thoughts, not intending to contest the presented discussion, but just for the sake of curiosity.

Could the order of effectiveness be related to the stability of the chelate formed between Ca and the antiscalants (apart from the fact that a substoichiometric amount is being used)? If that is really the case, are you aware of any study or intend to test in the future some other salt that has an inverse order of chelate stability? At least for HEDP, the stability of complexes is directly related to the pH. In that case, could the water splitting arising from the CEM that locally reduces the pH at the CEM surface (as depicted in Figure 3) impair the sequestration of Ca by HEDP and thus reduce its effectiveness as discussed in lines 307-327? 

Author Response

Thank you vey much for a different look at our results. The corresponding text is added at the end of manuscript, see attachment.

Reviewer 3 Report

Submitted paper describes the application of two fluorescent-tagged antiscalants in electrodialysis proccessing of solutions prone to gypsum scaling. By the use of gypsum deposits and antiscalant visualization authors reveal mechanisms of scale inhibition and explain the differences between two types of antiscalants. For the examination, different types of visualisation techniques were used - SEM, EDS, optical and fluorescent microscopy.

Paper is well prepared. My only recommendation is to broaden the literature review in Introduction. For example, there are articles about pulse electric fields (page 2, line 63) in electrodialysis which are newer and were not mentioned in text, e.g.:

Dufton, G. et al, Positive Impact of Pulsed Electric Field on Lactic Acid Removal, Demineralization and Membrane Scaling during Acid Whey Electrodialysis, doi: 10.3390/ijms20040797

Sosa-Fernandez, P. A. et al.,  Improving the performance of polymer-flooding produced water electrodialysis through the application of pulsed electric field, doi: 10.1016/j.desal.2020.114424

Haddad, M. et al, Eco-efficient treatment of ion exchange spent brine via electrodialysis to recover NaCl and minimize waste disposal, doi: 10.1016/j.scitotenv.2019.06.539

Articles about the use of antiscalants in electrodialysis (page 2, line 75) - I agree there are only few articles about this topic, but, actually, that is the reason why the literature review in this field should be precise. I found at least three more papers:

Hanrahan, C. et al., High-recovery electrodialysis reversal for the desalination of inland brackish waters, doi: 10.1080/19443994.2015.1041162

Xu, X. et al., Study of polyethyleneimine coating on membrane permselectivity and desalination performance during pilot-scale electrodialysis of reverse osmosis concentrate, doi: 10.1016/j.seppur.2018.06.070

Melnyk, L. A., Removal of Mn(II) compounds from water in electrodialysis desalination, doi: 10.3103/S1063455X15030042

Materials and methods are well described and clearly explained, results are presented in detail and reasonably discussed. 

I recommend to publish this paper after the extension of some parts of introduction as I mentioned in previous paragraphs.

Author Response

We would like to thank the reviewer for the positive feedback and appreciate reviewer’s time and effort to improve the quality of the article.

In accordance with the Reviewer advice, the introduction has been complemented by the articles kindly suggested by the Reviewer. However, in our opinion, one of the articles (Sosa-Fernandez, P. A. et al.) is not the best example of PEF application for fouling mitigation, since the authors state that the current mode doesn’t affect the fouling process, but energy consumption and demineralization percentage. On the whole, we highly appreciate the Reviewer advice which is very useful and helped to improve the quality of our paper. The changes in the reference list are highlighted in yellow, see attachment.
